# DIFFERENCE-MASKING:
## Choosing What to Mask in Continued Pretraining

**Alex Wilf**[*], **Syeda Nahida Akter**[*], **Leena Mathur, Paul Pu Liang,**
**Sheryl Mathew, Mengrou Shou, Eric Nyberg, Louis-Philippe Morency**
Carnegie Mellon University
{awilf,sakter}@cs.cmu.edu

## Abstract

The self-supervised objective of masking-and-predicting has led to promising performance gains on a variety of downstream tasks. However, while most approaches randomly mask tokens, there is strong intuition that deciding *what to mask* can substantially improve learning outcomes. We investigate this in continued pretraining setting in which pretrained models continue to pretrain on domain-specific data before performing some downstream task. We introduce DIFFERENCE-MASKING, a masking strategy that automatically chooses what to mask during continued pretraining by considering what makes a task domain *different* from the pretraining domain. Empirically, we find that DIFFERENCE-MASKING outperforms baselines on continued pretraining settings across four diverse language-only and multimodal video tasks.

## 1 Introduction

Inspired by the distributional hypothesis in the language domain (Harris, 1954), masking is a self-supervised learning (SSL) objective in which a model attempts to reconstruct hidden portions of data from the surrounding context. Masking has enabled breakthrough performance on tasks from a variety of domains, such as language, vision, and speech (Devlin et al., 2019; Li et al., 2021; Hsu et al., 2021; Ericsson et al., 2022), motivating interest in researching how masking strategies influence representation learning in SSL.

Masked prediction has recently been applied to adapt pretrained models to specific downstream tasks by continuing to pretrain models on in-domain unlabelled data (Dery et al., 2023). Masking in this continued pretraining setting been shown to be particularly effective when the target domain differs substantially from the pretraining domain (Gururangan et al., 2020).

While prior work has studied how the *amount masked* influences model learning (He et al., 2022),

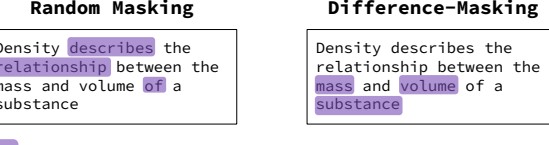

Figure 1: DIFFERENCE-MASKING automatically selects *what to mask* based on what makes the task domain *different* from the pretraining domain, enhancing model learning on the end task.

most masking approaches randomly choose which parts of the data to mask. Although it is understudied in SSL, deciding *what to mask* is a critical component in human education (Pajares and Miller, 1997; Bjork and Linn, 2006). Educators designing "fill-in-the-blank" assessments for students must decide what content to mask in order to effectively assess student understanding of a domain (Bae and Lee, 2018). For example, in a real-world "fill-in-the-blank" chemistry test, a teacher might choose to mask domain-specific words ("density", "silicon") to assess student learning, instead of masking domain-irrelevant words ("example", "process").

In this paper, we propose DIFFERENCE-MASKING, a novel approach for automatically selecting *what to mask* during continued pretraining. Our strategy first identifies *anchors* that describe what makes a target domain different from the pretraining domain and then determines what to mask during continued pretraining based on similarity to those anchors.

In experiments spanning four diverse language-only and multimodal video datasets (ACL-ARC, ChemProt, TVQA, and Social-IQ), we find that DIFFERENCE-MASKING outperforms strong baselines, supporting our hypothesis that *masking based on what is different* about a task provides strong representation for continued pretraining. We provide intuitions to explain the strong performance of DIFFERENCE-MASKING, along with extensive

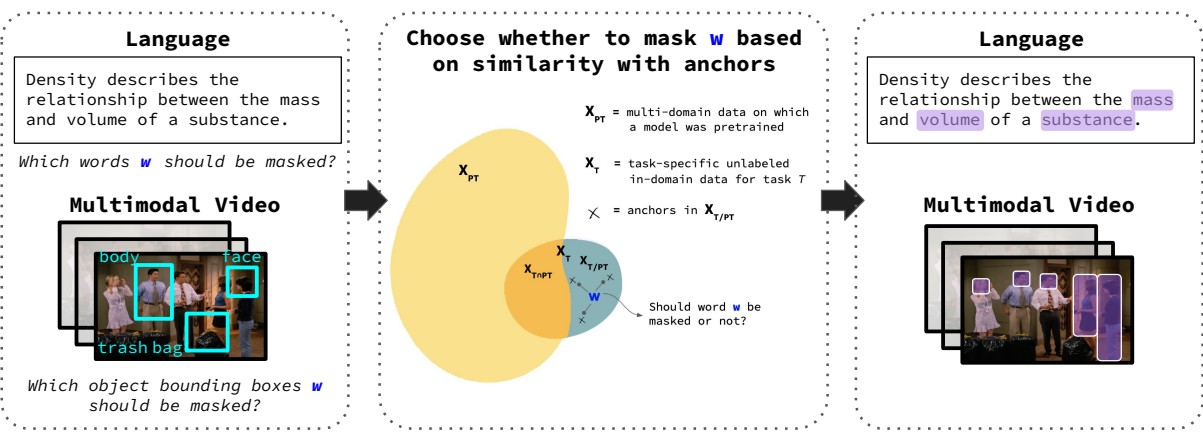

Figure 2: DIFFERENCE-MASKING: an approach to choosing what to mask during continued pretraining that prioritizes masking concepts that make the target domain different from the pretraining domain. DIFFERENCE-MASKING does this by first selecting *anchor topics* relating to the downstream task, and then by masking words or bounding boxes based on their similarity to those anchor topics.

analyses and ablations to better understand the performance of our method. Our code is publicly available.

## 2 Related Work

Masking relies on the distributional hypothesis, which posits that the meaning of a word can be inferred from its context (Harris, 1954). Masking in NLP has functioned as an effective SSL strategy when training models such as BERT (Devlin et al., 2019) and XL-Net (Yang et al., 2019). Although random masking has been more closely studied in NLP than non-random masking, there are three closely related works to ours from NLP.

EntityBERT (Lin et al., 2021) masks tokens based on whether they are part of "entities" recognized by a domain-specific pretrained named-entity-recognizer. Salient Span Masking (SSM) (Guu et al., 2020) is a similar method that uses a named-entity-recognition model to mask out a single entity for the downstream task of open-domain QA. However, these approaches require a domain-specific pretrained entity-tagger, and the masking strategy they determine is the same for any domain to which that tagger is applied. In contrast, DIFFERENCE-MASKING determines what to mask without pretrained entity-taggers, and its masking strategy can change depending on the unlabelled data in the task domain.

Selective Masking (Gu et al., 2020) uses data from the downstream task to decide which tokens to mask during continued pretraining by estimating how much each token contributes to improved downstream task performance. It is important to note that Selective Masking uses supervised downstream task labels, whereas DIFFERENCE-MASKING is entirely self-supervised.

Prior work from the vision community has also contributed to an understanding of masking strategies, primarily by using the attention of the model during SSL training to determine what to mask. MST (Li et al., 2021) uses attention maps to determine "non-essential regions" to mask, while AtnMask (Kakogeorgiou et al., 2022) does the opposite by masking the most attended-to regions. Unlike DIFFERENCE-MASKING, these approaches do not take into account domain-specific information when determining their masking strategy. This can be an impediment to performance when the model's attentions do not already contain information about what is important in a given input sequence.

## 3 DIFFERENCE-MASKING

This section describes the motivation and implementation of DIFFERENCE-MASKING: our self-supervised method to determine a masking strategy for continued pretraining. The overall process is depicted visually in Figure 2.

### 3.1 Problem Setting

We are given a model which has been pretrained on multi-domain data drawn from domain distribution $X_{PT}$ (e.g., a model such as RoBERTa pretrained

on a large multi-domain corpus). We are interested in how to adapt this pretrained model to a specific target domain $X_T$ without observing task labels $Y$.

Continuing pretraining on $X_T$ has emerged as a popular solution approach to this problem (Gururangan et al., 2020; Dery et al., 2023).

## 3.2 Motivation and Notation

If the masking objective is used to train models to learn word representations (Harris, 1954; Devlin et al., 2019), a natural question emerges: which words is it most important that our models learn to represent? We believe that this question may be important to effectively continue pretraining on specialized domains. We expect that continued pretraining can benefit from a masking strategy that considers what makes a task-domain different.

This leads to the intuition behind DIFFERENCE-MASKING: to train on what makes a target domain different from the pretraining domain. For example, in a corpus about chemistry we would expect that the task of masking and predicting words strongly related to chemistry such as "molecule" will lead to better learning outcomes than words such as "analysis", which could be related to chemistry in addition to many other domains.

Formally, we term $X_{T \cap PT}$ as the concepts likely to appear in both $X_T$ and $X_{PT}$ (e.g., "analysis"), and we term $X_{T/PT}$ as the concepts that make the domain $X_T$ different from $X_{PT}$ (e.g., "molecule"). With this notation, we can now express our intuition in terms of mutual information with the downstream task $Y$: we intuit that concepts common in $X_T$ but uncommon in $X_{PT}$ (i.e., in $X_{T/PT}$) share higher mutual information with the task label than concepts found in both domains ($X_{T \cap PT}$) do:

$$I(X_{T/PT}; Y) > I(X_{T \cap PT}; Y) \qquad (1)$$

The goal of DIFFERENCE-MASKING then is to learn representations during masking that capture the information unique to the domain ($X_{T/PT}$) which is more relevant for the downstream task.

## 3.3 Our Approach: DIFFERENCE-MASKING

To learn masked representations that capture the information unique to the domain ($X_{T/PT}$), our proposed DIFFERENCE-MASKING approach proceeds in two steps:

1. **Finding difference anchors**: We first determine which words are most commonly found in domain $X_T$ and *not* commonly found in

general domains $X_{PT}$. We term these words **difference anchors** that summarize the concepts unique to $X_T$.

2. **Masking based on differences**: Using these difference anchors, we determine the likelihood that each word should be masked based on its similarity to these difference anchors. We sample from this probability distribution to decide what to mask during MLM continued pretraining.

These steps are explained in detail in the following subsections.

## 3.4 Finding Difference Anchors: TF-ICF

Our goal is to determine a set of corpus-level difference anchors that are representative of the differences between the pretraining domain $X_{PT}$ and the task domain $X_T$. Since our goal is to design a simple yet effective method for finding these differences, we use of a modified version of the widely used TF-IDF scoring function from the field of statistical NLP (Jones, 1972). TF-IDF determines the ratio of how frequently a word appears in a *document* compared to how frequently the word appears in *other documents in a corpus*. Because we are attempting to find words that make a target *corpus* $X_T$ different from general pretraining *corpora* $X_{PT}$, the score of a word is highest when it appears frequently in our corpus ($X_T$) and infrequently in the multi-domain pretraining corpus ($X_{PT}$). We denote our approach as **TF-ICF** for term-frequency, inverse-*corpus*-frequency, expressed by the following scoring function:

$$\text{TF-ICF}(w_i) = \frac{\text{freq}(w_i, X_T)}{\text{freq}(w_i, X_{PT})} \qquad (2)$$

To effectively capture word frequencies in the general distribution of the English Language used for pretraining ($X_{PT}$), we use unigram counts derived from the Google Web Trillion Word Corpus (Brants and Franz, 2006; Norvig, 2009).

We score all words in $X_T$ with this metric and choose the top $K$ as anchors $A$ to represent the domain, where $K$ is a hyperparameter of our method. We analyze the impact of this hyperparameter in Section 5.3.

## 3.5 Masking Based on Differences

DIFFERENCE-MASKING then masks words based on similarity to anchors $A$. Formally, we define

similarity between a word $w$ and an anchor word $A_k$ as the cosine similarity of the words' BERT (Devlin et al., 2019) embeddings.

$$\text{sim}(w, A_k) = \cos(\text{BERT}(w), \text{BERT}(A_k)) \quad (3)$$

In order to choose words to mask, we generate probability distribution $\alpha$ over the words in the sentence $x$ to represent the probability that each word should be masked. We determine the weight $\alpha_i$ of each word $w_i$ by calculating its similarity score with the *most similar* anchor word in $A$ (we explore other strategies in our experiments). This value is normalized over the length of the sequence to ensure the probability distribution sums to 1.

$$\alpha(w_i) = \frac{\max_{k \in K} \text{sim}(w_i, A_k)}{\sum_{j=1}^{N} \max_{k \in K} \text{sim}(w_j, A_k)} \quad (4)$$

DIFFERENCE-MASKING then masks terms by sampling from distribution $\alpha$ without replacement, and the model attempts to reconstruct the masked terms from the surrounding context.

**Multimodal Implemention of DIFFERENCE-MASKING** To apply our method to the visual domain, we draw on work from the vision community in which visual representations are grouped at the object level (Baradel et al., 2018; Sajjadi et al., 2022) and use object labels (e.g. person, car...etc) from a state-of-the-art object detector (Wang et al., 2021; Zhang et al., 2016) to calculate similarity with the anchor words. A detailed description of our implementation of DIFFERENCE-MASKING in the multimodal setting can be found in Appendix B.

## 4 Experimental Settings

Our experiments evaluate whether DIFFERENCE-MASKING's masking strategy leads to performance improvements on challenging language-only and multimodal video understanding tasks. We follow the experimental setting from (Gururangan et al., 2020), in which unlabelled data from the downstream task domain is used for continued pretraining before eventually performing downstream task finetuning. This is a popular SSL setting because it represents a computationally-feasible way to test the effectiveness of self-supervised representation learning methods (e.g. without recreating a pretrained model), and it is realistic to modern approaches which rely heavily on pretrained models (Dery et al., 2023).

Experiments are performed to allow each model to learn as long as needed during continued pretraining, only stopping when validation error increases (early-stopping). Each result is averaged across five random seeds. Hyperparameter settings and data preprocessing details can be found in Appendix A.

### 4.1 Datasets

**Language-only Datasets** As in Gururangan et al. (2020); Dery et al. (2023), we conduct experiments with the **ChemProt** dataset (Kringelum et al., 2016), a relation classification task that uses chemistry documents. ChemProt is a low-resource classification task with a large amount of in-domain unlabeled data, making it a realistic setting in which SSL is helpful in continued pretraining.

We also conduct experiments with the **ACL-ARC** task (Jurgens et al., 2018), a citation intent task based on the ACL Anthology Reference Corpus (Bird et al., 2008) used in continued pretraining experiments in (Gururangan et al., 2020). We use train, validation, and test splits for both datasets from (Dery et al., 2023; Gururangan et al., 2020).

**Multimodal Datasets** We also experiment on continued pretraining for two challenging multimodal video understanding tasks. **TVQA** (Lei et al., 2018) is a dataset containing 21,792 videos from 6 American television shows and questions and answers related to the videos. Each question is paired with 5 answer choices (one correct answer and 4 incorrect answers), and corresponding video, audio, and subtitles.

**Social-IQ** (Zadeh et al., 2019) contains 1,250 videos of social situations and questions and answers pertaining to the videos. Each question has corresponding video, audio, and subtitles. We use the train, validation, and test splits from the publicly available datasets.

We use performance metrics consistent with prior work (Gururangan et al., 2020; Dery et al., 2023): F1 score for ACL-ARC and classification accuracy for ChemProt, TVQA, and Social-IQ.

### 4.2 Baseline Methods

**Random Masking** Most masking approaches choose tokens or words to mask with a uniform random probability (Devlin et al., 2019; Yang et al., 2019). We consider both the token-level and word-level approaches in our experiments. Formally, the probability $\alpha_i$ that word or token $x_i$ in a sequence

| Masking Strategy | Language-Only | | Multimodal | |
| --- | --- | --- | --- | --- |
| | ACL-ARC | ChemProt | Social-IQ | TVQA |
| Random Masking (Word) | $62.05_{2.21}$ | $81.90_{0.51}$ | - | - |
| Random Masking (Token) | $63.74_{1.97}$ | $82.82_{0.23}$ | $69.05_{0.52}$ | $73.75_{0.31}$ |
| MST (Li et al., 2021) | $65.61_{0.13}$ | $83.17_{0.17}$ | $68.37_{0.49}$ | $81.14_{0.30}$ |
| AttnMask (Kakogeorgiou et al., 2022) | $66.30_{1.67}$ | $83.53_{0.56}$ | $70.18_{0.71}$ | $81.57_{0.12}$ |
| DGA (Ke et al., 2023) | $67.20_{0.27}$ | $70.67_{0.30}$ | - | - |
| Selective Masking (Gu et al., 2020) | $69.06_{1.80}$ | $82.94_{0.47}$ | - | - |
| EntityBERT (Lin et al., 2021) | $71.09_{0.25}$ | $82.04_{0.40}$ | - | - |
| Salient Span (Cole et al., 2023) | $71.94_{0.58}$ | $82.41_{0.21}$ | - | - |
| DIFFERENCE-MASKING | $\mathbf{74.04}_{2.01}$ | $\mathbf{83.94}_{0.39}$ | $\mathbf{71.37}_{0.58}$ | $\mathbf{81.73}_{1.13}$ |

Table 1: We find that DIFFERENCE-MASKING outperforms strong baselines in both the language and multimodal experimental settings. We note that our entirely self-supervised method also outperforms Selective Masking, which uses labelled data to inform its masking strategy. Values are average results over five trials, subscripts are standard deviations.

of length $N$ will be masked in random-masking is

$$\alpha_i = \frac{1}{N} \quad (5)$$

**AttnMask (Kakogeorgiou et al., 2022)** is a *domain-agnostic* token-based masking approach in which the likelihood of masking a given token is proportional to how attended-to that token is by the [CLS] token, averaged across the different heads of the transformer. Formally, this approach can be seen as defining a function $g_{att}$ which takes in model $f_\theta$, sequence of tokens $x$, and index $i$ and outputs how attended-to token $x_i$ is.

$$\alpha_i \propto g_{att}(f_\theta, x, i) \quad (6)$$

**MST (Li et al., 2021)** is an approach very similar to AttnMask, except that it masks "non-essential regions", effectively corresponding to an inverse weighting based on the attention of the model to the token $x_i$.

$$\alpha_i \propto g_{att}(f_\theta, x, i)^{-1} \quad (7)$$

**Selective Masking (Gu et al., 2020)** chooses tokens to mask based on whether adding each token will improve downstream task accuracy as measured by the difference between the downstream task performance when using the full sequence $x$ versus using only the sequence up to and including the token $x_i$. Notably, this approach *uses downstream task labels to guide the choice of mask* in continued pretraining, whereas DIFFERENCE-MASKING *is self-supervised*.

$$\alpha_i \propto P(y \mid x) - P(y \mid x_{[:i]}) \quad (8)$$

**DGA (Ke et al., 2023)** is another relevant work that proposes a masking strategy for NLP model adaptation. However, unlike the methods described above, DGA chooses which *attention heads* to mask instead of choosing which *tokens* to mask, assigning importance to attention heads based on the gradient of the loss between the model's representations of two differently-masked versions of the same input. Additionally, DGA encourages the model to learn integrated representations of the target domain and general knowledge using a contrastive loss.

**EntityBERT (Lin et al., 2021)** masks tokens based on whether they are part of "entities", as defined by a domain-specific named-entity-recognition (NER) model. The original paper uses the PubMedBERT model, trained originally on the clinical domain. We also implement **Salient Span Masking** (Guu et al., 2020), which in this case is the same as the EntityBERT approach applied only to mask a single word in the sentence. To apply the approach to the ChemProt and ACL-ARC domains requires NER models effective in those domains. For ChemProt we used the BioBERT model (Lee et al., 2019) fine-tuned in NER task with BC5CDR-chemicals (Li et al., 2016) and the BC4CHEMD (Krallinger et al., 2015) corpus and for ACL-ARC we used the popular SciBERT model (Beltagy et al., 2019).

### 4.3 Experimental Methodology

**Language-only** We reproduce the experimental setting from AANG (Dery et al., 2023), which employs a pretrained 110M RoBERTa$_{base}$ model with

two heads: one for continued pretraining and one for the downstream task. Our hyperparameters and other detailed configuration notes are described in Appendix A.

**Multimodal**   We conduct our multimodal experiments using a strong pretrained model: MERLOT-Reserve (Zellers et al., 2022), a large multimodal transformer pretrained with a contrastive multimodal prediction objective on a dataset of 20 million Youtube videos.

To experiment with masking strategies in the multimodal setting, we continually pretrain a 200M MERLOT-Reserve$_{base}$ model by masking-and-predicting visual patches. We evaluate the learned representation quality by freezing the model and finetuning only the linear classifier layer on the downstream task following (Wilf et al., 2023)'s methodology.

A detailed description of our implementation of DIFFERENCE-MASKING in the multimodal setting can be found in Appendix B, and our hyperparameters can be found in Appendix A.

## 5   Results and Discussion

### 5.1   Comparison with Baseline Approaches

Our experiments compare our proposed DIFFERENCE-MASKING with established baselines including Random Masking (at the word and token level), AttnMask (Kakogeorgiou et al., 2022), MST (Li et al., 2021), Selective Masking (Gu et al., 2020), DGA (Ke et al., 2023), EntityBERT (Lin et al., 2021), and Salient Span Masking (Cole et al., 2023). The results are summarized in Table 1. We find that DIFFERENCE-MASKING shows strong results compared to baselines across language-only and multimodal video understanding tasks.

Notably, our approach demonstrated superior performance on the ACL-ARC dataset with an accuracy of 74.04%, a marked improvement over the random token baseline (63.74%) and a substantial improvement over the best baseline (Salient Span Masking, 71.94%). Our approach also surpassed Selective Masking (69.06%). This is surprising because Selective Masking uses downstream task labels to inform its masking strategy whereas DIFFERENCE-MASKING is self-supervised.

Results on the ChemProt dataset are also encouraging, showing that DIFFERENCE-MASKING achieves an accuracy of 83.94%, marginally better

than all the baselines, including Random Masking (82.82%), AttnMask (83.53%), and EntityBERT (82.04%). Similarly to Selective Masking, the EntityBERT and DGA masking strategies were originally tested on much larger datasets, which may suggest a limitation of these methods in the low-resource continued pretraining setting.

DIFFERENCE-MASKING also demonstrates robust performance in multimodal settings. On the Social-IQ dataset, DIFFERENCE-MASKING achieved an accuracy of 71.37%, outperforming the Random Masking (69.05%), AttnMask (70.18%), and MST (68.37%) methods. We were unable to compare our approach with Selective Masking and EntityBERT on these datasets due to the language-only design of their entity taggers. In contrast, our method is not limited to the language domain, and, in fact, performs well in the multimodal setting. And on the TVQA dataset, DIFFERENCE-MASKING achieved an accuracy of 81.73%, outperforming the Random Masking approach substantially (73.75%) and the AttnMask approach marginally (81.57%).

These results highlight the effectiveness and versatility of the DIFFERENCE-MASKING approach across various language and multimodal datasets.

### 5.2   What is masked?

In this section, we investigate what is masked by DIFFERENCE-MASKING and its link to downstream task performance.

On the ACL-ARC task, we find that the most frequently masked words in the ACL-ARC task had an interesting grounding in human intuition. The ACL-ARC task is a citation intent task on a corpus comprising ACL papers. As the subject of ACL papers can vary widely, comprising multiple sub-domains and research fields, we were curious how DIFFERENCE-MASKING's masking strategy would handle this domain.

We found that the most frequently masked words *closely-aligned with the ACL paper submission tracks* describing the high-level topic categories for papers. For example, some of the most frequently masked words were "learning", "information", "translation", "semantic", and "lexical". These words closely correspond to submission tracks "Machine Learning for NLP", "Information Extraction", "Machine Translation", and "Semantics: Lexical". Since submission tracks for ACL can be seen as a set of topics that span the space

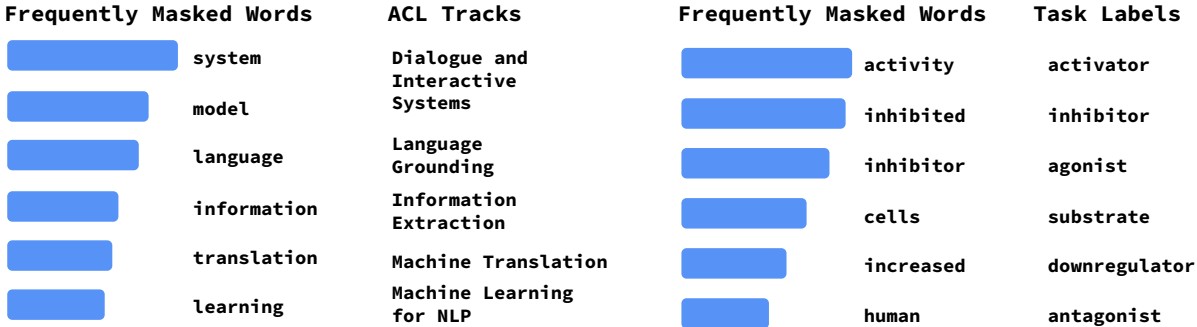

Figure 3: The most frequently masked words chosen by the DIFFERENCE-MASKING algorithm across the ChemProt and ACL-ARC tasks. We find that for the ChemProt dataset, the masks we find automatically through unlabelled data partially recover the end task labels.

of ACL papers, this supports our hypothesis that masked words chosen by DIFFERENCE-MASKING align with what *makes this domain different*.

On the ChemProt task we also found an interesting pattern in what was masked. The objective of the ChemProt task is to determine a type of relation corresponding to a type of biochemical interaction between entities in the text, where labels include words such as "activation", "inhibitor", and "antagonist". Interestingly, we find that some of the words DIFFERENCE-MASKING chooses to mask most often are *the same words as the labels for the downstream task*. This result is also visualized in **Figure 3**. Some of the most-masked words by DIFFERENCE-MASKING are "activity", followed by "inhibited", "inhibitor", and "antagonist". This is a fascinating result because it suggests that *in masking what makes the ChemProt domain unique*, DIFFERENCE-MASKING is determining a self-supervised objective that is *highly similar to the downstream task* without accessing the downstream task labels.

In the multimodal setting we also find an interesting grounding of how DIFFERENCE-MASKING chooses masks in human intuition. Reasoning about social interactions is believed by many psychologists to rely heavily on understanding visual body language cues (De Stefani and De Marco, 2019; Keck et al., 2022). Social-IQ is designed to test these kind of social intelligence capabilities with subtle questions such as "How do the men in the room feel about each other?" and "Do the people in this video feel comfortable about the clown being there?". In contrast, TVQA tests more general video understanding with question and answer

| Method | TVQA | Social-IQ |
|---|---|---|
| Random | 17% | 15% |
| AttnMask | 38% | 19% |
| DIFFERENCE-MASKING | 40% | 90% |

Table 2: For each method, we analyze what percent of tokens are chosen to be masked from within bounding boxes over people as opposed to objects.

types including those that target visual reasoning about non-human entities and non-visual reasoning from specifically text or audio modalities.

As such, we would expect that our continued pre-training strategy would choose to prioritize masking tokens representing human body language more often in Social-IQ than in TVQA. We found that this was in fact the case. Interestingly, we found that AttnMask baseline also picked up on a similar trend in its attempt to mask based on where attention already focuses, although the trend is much more pronounced in our approach.

The findings in **Table 2** demonstrate that DIFFERENCE-MASKING chooses to mask substantially fewer visual tokens corresponding to people than to objects in TVQA, (40%) in comparison to Social-IQ (90%). On the Social-IQ dataset, where the performance difference is more pronounced over the closest baseline ($\uparrow$ 1.76% over AttnMask), the difference between the proportion of tokens masked from people by these approaches is also most pronounced (90% in DIFFERENCE-MASKING vs 19% in AttnMask).

## 5.3 Sensitivity Analysis

**Similarity Function** As described in Section 3, DIFFERENCE-MASKING determines masking probabilities by comparing the anchor representations to the token representation. Because the token representation is a single vector and the anchors are a group of vectors, similarity can be defined multiple ways. Table 1 shows results from the "nearest-neighbor" approach to determining similarity described in Section 3.5, motivated by the intuition that a domain can have many *sub-domains* and if a token is close to *any* one of these concepts it should be prioritized for masking. For example, the ACL-ARC corpus has many sub-domains, including the over twenty different submission tracks described in Section 5.2. If a paper is about linguistics, it may be important to mask words similar to "language", whereas if a paper is heavy on ML theory, another anchor might be more appropriate to mask in order to best understand the work.

An alternative approach that could be to determine scores by relation to the centroid of the anchor embeddings: in essence, determining whether the token in question is similar to the *anchors on aggregate*. We would expect that this approach would perform similarly to ours on a narrowly-defined dataset such as ChemProt, but substantially differently on a multi-domain dataset such as ACL-ARC. We evaluate this alternative in Table 3.

|  | ACL-ARC | ChemProt |
|---|---|---|
| Centroid | 69.02 | 83.66 |
| Nearest-Neighbor | **74.04** | **83.94** |

Table 3: Ablating DIFFERENCE-MASKING's anchor-scoring function based on nearest-neighbor and replacing it with one based on similarity with the anchor embeddings' centroids leads to performance degradation. This provides evidence for our hypothesis that the nearest-neighbor scoring function helps make DIFFERENCE-MASKING robust to anchor selections.

We find that the nearest-neighbor strategy does, in fact, outperform the centroid strategy, especially on the ACL-ARC task. This supports our intuition that the nearest-neighbor strategy is particularly helpful when there is a complex or peaky domain.

**Number of Anchors** In considering the relationship between the anchors and the downstream task, we also investigate how the choice of the number of anchors ($K$) impacts the downstream performance. We expect that too few anchors will not be expres-

sive enough to determine a strong masking strategy, and too many anchors may begin to overfit to niche concepts that are not representative of the domain. We find that there is indeed a "sweet spot", and interestingly that it is the same for both datasets: 20. These results are visualized in Figure 4.

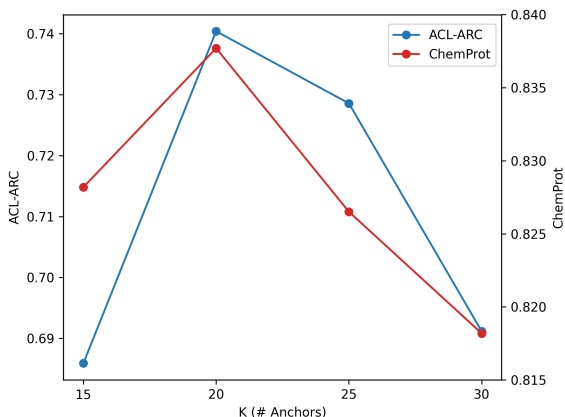

Figure 4: Performance on both tasks is best at the hyperparameter $K = 20$ anchors. We hypothesize that each task may have an optimal setting of this hyperparameter.

## 6 Conclusion

In this paper we introduce DIFFERENCE-MASKING, a method for identifying what makes a target domain unique and using this information to guide a strategy that chooses *what to mask* during SSL continued pretraining. We find that our method outperforms strong baselines across diverse language and multimodal video understanding tasks. We provide a detailed discussion of *what is masked* in DIFFERENCE-MASKING and why our method performs well on various tasks. The cross-task applicability of DIFFERENCE-MASKING supports the effectiveness of our framework for SSL pretraining in language, vision, and other domains.

## 7 Limitations

As described in Section 3, DIFFERENCE-MASKING is based on the intuition that it is more beneficial to mask based on what is unique ($X_{T/PT}$) about a downstream task's domain. However, it is challenging to find what makes a domain unique; therefore, our method is an approximation of $X_{T/PT}$. We believe future work may find it fruitful to investigate additional methods for approximating this, including modifications on the TF-ICF method we proposed. In Section 5, we provided intuition, empirical results, and analysis to

understand why our method outperformed attention masking baselines by a larger margin on Social-IQ than on TVQA. A broader investigation of why DIFFERENCE-MASKING during pretraining is beneficial by a larger margin to some downstream tasks than to others would be helpful to the community.

## 8 Ethics Statement

We believe that self-supervised learning is a promising direction for the machine learning community. This does not discount the salient arguments made about the social and enviromental risks of large models (Bender et al., 2021; Strubell et al., 2019). We believe that works such as ours, which study SSL in a resource-constrained context, both increase access to those with limited compute resources and conform to a more environmentally-sustainable way of doing research.

## Acknowledgements

This material is based upon work partially supported by National Science Foundation awards 1722822 and 1750439, and National Institutes of Health awards R01MH125740, R01MH132225, R01MH096951 and R21MH130767. Any opinions, findings, conclusions, or recommendations expressed in this material are those of the author(s) and do not necessarily reflect the views of the sponsors, and no official endorsement should be inferred.

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

## A  Detailed Experimental Settings

In this section, we provide an overview of the experimental conditions utilized in our study. To ensure fair comparisons with our baselines, we maintain a consistent set of hyperparameters for both continuous pretraining and fine-tuning. For language tasks, we largely adhere to the hyperparameters employed in (Gururangan et al., 2020). Throughout our experiments, we maintain a masking ratio of 25% in both language and multimodal settings. We adopt a static masking strategy, replacing masked tokens with random values.

| Hyperparameters | CPT | | FT | |
|---|---|---|---|---|
| | Language | Multimodal | Language | Multimodal |
| learning_rate | 0.0001 | 0.000005 | 1.00E-06 | 5.00E-06 |
| num_train_epochs | 150 | 20 | 10 | 20 |
| eval_every_n_epochs | 30 | 1 | 1 | 1 |
| patience | 20 | 5 | 3 | 5 |

Table 4: List of hyperparameters used in both continuous pretraining (CPT) and finetuning (FT).

We reproduce MERLOT-Reserve's original training on TVQA: we decompose samples in Social-IQ and TVQA from the form (Question, All Answers, Video Information) into a list of 3-tuples: (Question, Candidate Answer, Video Information). MERLOT scores each candidate answer independently, given the question and video, and is trained with loss that encourages the model to minimize estimated likelihood of incorrect answers and maximize likelihood of correct answers.

From video frames, we mask image patches into 16x16 patches as determined by MERLOT-Reserve's backbone image transformer ViT (Dosovitskiy et al., 2021). The language experiments took nine hours of runtime each on a single 12GB GPU, and the multimodal vision experiments required six hours on a single TPU v2-8.

## B  Masking Video Tokens

Following the intuition from language, we hypothesize that masking and predicting small patches of an image may be testing *local* capabilities (e.g. determining what an eye looks like from the rest of the face) rather than *global* capabilities (e.g. determining what a person's face looks like from the rest of the scene, including other people's faces).

Accordingly, instead of masking low-level image patches, we mask groups of patches corresponding to a higher level semantic entity: bounding boxes over objects in the image. We see this approach as a visual analogue for masking at the word-level

instead of the token-level in our language experiments. We found that $K = 1$ performed much better than other values, where the selected anchor word was "person". We considered two possible bounding boxes associated with people: bounding boxes over faces and bodies. We evaluated both options and found that considering entire bounding boxes over people's bodies (including their faces) performed the best. These results are shown in Table 5.

| Masking Strategy | TVQA | Social-IQ |
|---|---|---|
| Random Masking | 73.75 | 69.05 |
| DIFFERENCE-MASKING (Face) | 81.51 | 69.13 |
| DIFFERENCE-MASKING (Body) | **81.73** | **71.37** |

Table 5: Results of DIFFERENCE-MASKING on multimodal video understanding benchmarks TVQA and Social IQ. DIFFERENCE-MASKING leads to an improvement of 8% and 2% accuracy over random accuracy.

We extracted body detection coordinates using UniTrack (Wang et al., 2021) and face detection coordinates using MTCNN (Zhang et al., 2016).

Apart from the bounding box strategy, we also experimented with masking patches chosen by differences between CLIP embeddings (Radford et al., 2021) of the anchor and the vision patch directly (without bounding box labels). Our experiments validate that the CLIP-based masking strategy performs poorly compared to our bounding box strategy. One possible reason can be that CLIP is not robust enough for video datasets which led to masking patches that are not relevant to the anchor word "person".

| | TVQA | Social-IQ |
|---|---|---|
| CLIP (Radford et al., 2021) | 73.58 | 68.75 |
| DIFFERENCE-MASKING | **81.73** | **71.37** |

Table 6: We validate our hypothesis that masking patches using DIFFERENCE-MASKING is more effective than masking using CLIP similarity.

## C  Masking Language Tokens

In Section 4.3 we describe the motivation for using a word-level strategy in our implementation of DIFFERENCE-MASKING. An alternative implementation could be to assign each token in a word the same masking likelihood, and mask tokens only by this probability. This could result in some tokens

from the same word being masked where others are not. Our intuition is that for specialized domains such as chemistry, subword tokens may be trivial to predict from their neighbors, but whole words may not be trivial to predict given the context. For example, a word such as "phosphates" would be tokenized into "phos" and "-phates". We expect that it may be trivial to predict "phos" given "-phates" or vice versa, but it may be hard (and may promote a better understanding of the task) to predict the word "phosphates" given the context.

Empirically, we find that this decision improved performance substantially, as shown in the results in Table 7 below.

|  | ACL-ARC | ChemProt |
|---|---|---|
| Token | 0.6501 | 0.8224 |
| Word | **0.7404** | **0.8394** |

Table 7: We validate our hypothesis that masking tokens using DIFFERENCE-MASKING at the word-level is more effective than masking at the token-level.

