# OpenReview forum: "Difference-Masking: Choosing What to Mask in Continued Pretraining"
_EMNLP/2023/Conference — EMNLP 2023 Findings_

### Official Review · Reviewer_o2x7 · 2023-08-03

**Soundness:** 2

**Excitement:**

1: Poor: I cannot identify the contributions of this paper, or I believe the claims are not sufficiently backed up by evidence. I would fight to have it rejected.

**Paper Topic And Main Contributions:**

This paper studies the continued pretraining setting where pretrained models continue to pretrain on domain-specific data before performing some downstream task. The paper introduces a masking strategy that automatically chooses what to mask during continued pretraining by considering what makes a task domain different from pretraining domain. On continued pretraining settings the proposed method performs better across four diverse language-only and multimodal video tasks.

**Questions For The Authors:**

Even though this may be out of scope for this paper, I am wondering is continual pretraining needed? Especially for VL tasks including VideoQA I think the common setting is to finetune on the dataset using pretrained VL models.

**Reasons To Accept:**

The masking strategy is commonly used in VL and language model pretraining. Studying how to more effectively perform the masking is well-motivated and it is an important problem.

The continual pretraining setting is interesting.

Both language-only and multimodal tasks are evaluated.

**Reasons To Reject:**

Novelty of the proposed masking strategy is limited.

Only a 110M RoBERTabase model and a MERLOT-Reserve model are tested. It is hard to tell if the proposed method can be generalized to a more common scenario for example for large language models of billions of parameters.

In the multimodal setting, only the VideoQA is evaluated and for language-only setting, only ACL- ARC task is evaluated.

**Reproducibility:**

2: Would be hard pressed to reproduce the results. The contribution depends on data that are simply not available outside the author's institution or consortium; not enough details are provided.

**Reviewer Confidence:**

3: Pretty sure, but there's a chance I missed something. Although I have a good feel for this area in general, I did not carefully check the paper's details, e.g., the math, experimental design, or novelty.

---

> ### Author Rebuttal · Authors · 2023-08-29
>
> We sincerely appreciate your review and the insights provided about our work on continued pretraining. It's heartening to know that you recognize the importance of studying effective masking strategies, especially given its prevalence in VL and language model pretraining. We are glad that the significance of the continual pretraining setting and our exploration across both language-only and multimodal tasks resonated with you. We'll now proceed to address your concerns.
>
> --- "Novelty of the proposed masking strategy is limited." ---
>
> We respectfully disagree, but find this difficult to engage with because of a lack of explanation. Although tf-idf has been widely used in NLP, its application to determining what to mask has not been explored before (especially using the idea of tf-idf across corpuses, as we do in our paper). Additionally, our experiments and analyses show our method's usefulness in important application areas that may not be covered by standard pretraining. Underlying our use of tf-idf is the core idea to mask what makes a domain unique without labels, which is significantly different from the baselines we considered, which either mask randomly ([1]), by focusing on models' attentions ([2],[3]), by using a domain-specific entity tagger which requires pretraining, and does not "rank" masking probabilities by uniqueness ([4]), or by using downstream labels (which our method assumes we do not have access to) to influence the masking strategy ([5]).
>
> Perhaps this assessment of our novelty is due to our method's simplicity. Although Difference-Masking is relatively simple to implement, we do not believe this to be a weakness of the method – in fact quite the opposite. Because our method is simple to implement and operates in a setting with relatively small amounts of data, we think this may be an important baseline for future work that seeks to continually pretrain models on target domains – especially because it performs much more than complex baseline approaches on the low-resource setting.
>
> We hope this addresses R3's concern.
>
> --- "Only a 110M RoBERTabase model and a MERLOT-Reserve model are tested. It is hard to tell if the proposed method can be generalized to a more common scenario for example for large language models of billions of parameters." ---
>
> In this paper, we study how to determine which tokens to mask during continued pretraining. Encoder-based transformer models such as RoBERTa and MERLOT-Reserve randomly mask-and-predict a portion of the input tokens. We replace this random masking strategy with our proposed approach (as well as other non-random baselines) and find strong improvements. Decoder-based transformer models such as today’s multi-billion-parameter LLMs do not selectively mask-and-predict; they always predict the next token. Therefore, it is unclear how to apply “masking strategies” to a training paradigm that does not selectively mask (through random selection or otherwise).
>
> However, in response to this question, we attempted to do so to the best of our abilities by creating an instruction-tuning version of masked-language modeling with one open source LLM, Vicuna-7B. We found that Vicuna did actually perform slightly better with our method of selecting masks. Our experimental details and results are below:
>
> We conduct two stages of finetuning on pretrained Vicuna-7B using LoRA. To adapt autoregressive models in our setup, we design an instruction-tuning task for the first stage (CPT) where we replace some words (randomly vs Difference-Masking: TF-ICF) in the text (15%) with “%%%%” and ask the model to predict the replaced words. We only use the training set of the downstream tasks (ACL-ARC, Chemprot)  for CPT. In the second stage (FT), we finetune the CPT-ed model on the downstream tasks.
>
> Here is our exact hyperparameter setting: Batch size: 32, LR: 3e-4, Epochs: 3 (early stopping), LoRA: r=8, lora_alpha=16, lora_dropout=0.05, Schedule: Linear warmup with cosine decay, Warmup steps: 128, Precision: bfloat16.
>
> And the prompts we use for instruction tuning:
>
> CPT Prompts:
>
> prompts = f"Given a text from a scientific paper where some words are replaced by %%%%% , the assistant outputs those words in the text. \n\n Text: {text} \n\n Output:"
>
> FT Prompts:
>
> prompts = f"Given a text from a scientific paper which contains two phrases inside << >> and [[ ]], the assistant outputs the relations of those two phrases in the text \n\n Text: {text} \n\n Relation:"
>
> We run these experiments across five trials and find that Vicuna-7B performs slightly better on ACL-ARC (.5476) using Difference-Masking CPT as opposed to random (.5396).  On ChemProt too, our method performs slightly better: .8020 compared to .7982. The gap in these experiments may be narrower due to the fact that this kind of “masked language modeling” within an instruction tuning setup is not common for this class of models. This is our best attempt to simulate masking within these models.
>
> A deeper interpretation of this point, however, may be that it is not worthwhile to conduct research on masking strategies in encoder-based models such as RoBERTa and MERLOT in the era of LLMs such as the GPT models. To this point, it is worth mentioning that masking is particularly useful in other modalities right now, such as vision [9] and audio [10]. Understanding masking strategies may be useful for those modalities, if not for NLP at the moment. However, continued pretraining appears to be a relevant and meaningful research direction at the moment [1,11,13,14].
>
> --- "In the multimodal setting, only the VideoQA is evaluated and for language-only setting, only ACL-ARC task is evaluated." ---
>
> We would like to clarify some misconceptions about our experimental setting: it is incorrect that "for language-only setting, only ACL-ARC task is evaluated". We reference the ChemProt task, the other language-only dataset often in our paper, including in Table 1, Section 1, Section 4, Section 5.1, Section 5.2, Figure 3, Section 5.3, and Table 3, where we describe the dataset, our experimental setting, our findings, and detailed analysis involving the ChemProt task.
>
> It is true that both of our multimodal datasets (TVQA and Social-IQ) are video-based and use the question-answering paradigm for evaluation, but this is the case of many multimodal datasets and tasks ([6]). We test on four distinct datasets, two language-only and two multimodal video. We hope this clarifies any misconceptions about the nature of our experimental setting.
>
> --- Questions For The Authors: "Even though this may be out of scope for this paper, I am wondering is continual pretraining needed? Especially for VL tasks including VideoQA I think the common setting is to finetune on the dataset using pretrained VL models." —
>
> Continued pretraining has been shown to be particularly effective in settings where there is limited labelled data from the downstream task, and a larger amount of unlabelled in-domain data [1,13,14]. As labelled data can be costly to annotate (which is the case for VL tasks including VideoQA [12]), but unlabelled data is much easier to come by, it becomes increasingly important to have domain-adaptation strategies that can learn from larger amounts of in-domain unlabelled data. The limited label-setting is what makes the continued pretraining necessary. Our method is an approach to performing this continued pretraining better by determining what to mask based on the downstream domain. Continued pretraining can also be important in related settings applied to language and vision models, such as mitigating catastrophic forgetting [11].
>
> Regarding soundness and reproducibility, we carefully document our experimental setting and our method and describe the analyses and the baseline masking approaches in great detail. In the appendices, we provide our exact hyperparameter grid search applied to the original settings from Don't Stop Pretraining [1] and MERLOT-Reserve [7], as well as describe in detail how we implement masking at the bounding box and word level for our experiments. In addition, we have released our anonymized code and use publicly available dataset [8] for anyone to reproduce. We hope R3 will reconsider our soundness and reproducibility given these clarifications.
>
> --
>
> [1] Gururangan, Suchin, et al. "Don't stop pretraining: Adapt language models to domains and tasks." arXiv preprint arXiv:2004.10964 (2020).
>
> [2] Zhaowen Li, Zhiyang Chen, Fan Yang, Wei Li, Yousong Zhu, Chaoyang Zhao, Rui Deng, Liwei Wu, Rui Zhao, Ming Tang, et al. 2021. Mst: Masked self- supervised transformer for visual representation. Advances in Neural Information Processing Systems, 34:13165–13176.
>
> [3] Ioannis Kakogeorgiou, Spyros Gidaris, Bill Psomas, Yannis Avrithis, Andrei Bursuc, Konstantinos Karantzalos, and Nikos Komodakis. 2022. What to hide from your students: Attention-guided masked image modeling. In Computer Vision – ECCV 2022, pages 300–318. Springer Nature Switzerland.
>
> [4] Chen Lin, Timothy Miller, Dmitriy Dligach, Steven Bethard, and Guergana Savova. 2021. EntityBERT: Entity-centric masking strategy for model pretrain- ing for the clinical domain. In Proceedings of the 20th Workshop on Biomedical Language Processing, pages 191–201, Online. Association for Computa- tional Linguistics.
>
> [5] Yuxian Gu, Zhengyan Zhang, Xiaozhi Wang, Zhiyuan 686 Liu, and Maosong Sun. 2020. Train no evil: Selective 687 masking for task-guided pre-training. In Proceedings of the 2020 Conference on Empirical Methods in Natural Language Processing (EMNLP), pages 6966–6974, Online. Association for Computational Linguistics.
>
> [6] Baltrušaitis, Tadas, Chaitanya Ahuja, and Louis-Philippe Morency. "Multimodal machine learning: A survey and taxonomy." IEEE transactions on pattern analysis and machine intelligence 41.2 (2018): 423-443.
>
> [7] Zellers, Rowan, et al. "Merlot reserve: Neural script knowledge through vision and language and sound." Proceedings of the IEEE/CVF Conference on Computer Vision and Pattern Recognition. 2022.
>
> [8] https://anonymous.4open.science/r/Difference-Masking-842D/README.md
>
> [9] Peng, Zhiliang, et al. "A unified view of masked image modeling." arXiv preprint arXiv:2210.10615 (2022).
>
> [10] Niizumi, Daisuke, et al. "Masked spectrogram modeling using masked autoencoders for learning general-purpose audio representation." HEAR: Holistic Evaluation of Audio Representations. PMLR, 2022.
>
> [11] Cossu, Andrea, et al. "Continual pre-training mitigates forgetting in language and vision." arXiv preprint arXiv:2205.09357 (2022).
>
> [12] Vondrick, Carl, Donald Patterson, and Deva Ramanan. "Efficiently scaling up crowdsourced video annotation: A set of best practices for high quality, economical video labeling." International journal of computer vision 101 (2013): 184-204.
>
> [13] Dery, Lucio M., et al. "AANG: Automating Auxiliary Learning." arXiv preprint arXiv:2205.14082 (2022).
>
> [14] Gupta, Kshitij, et al. "Continual Pre-Training of Large Language Models: How to (re) warm your model?." arXiv preprint arXiv:2308.04014 (2023).

---

### Official Review · Reviewer_Yfzq · 2023-08-05

**Typos Grammar Style And Presentation Improvements:** See above
**Soundness:** 3

**Excitement:**

3: Ambivalent: It has merits (e.g., it reports state-of-the-art results, the idea is nice), but there are key weaknesses (e.g., it describes incremental work), and it can significantly benefit from another round of revision. However, I won't object to accepting it if my co-reviewers champion it.

**Paper Topic And Main Contributions:**

This paper improves continued pre-training, a.k.a. pre-finetuning, post-training or domain-adaptive pre-training by masking only the important/different words.

**Questions For The Authors:**

See above

**Reasons To Accept:**

1. Domain-adaptive pre-training is important.
2. The conventional masked language modeling based on random masks is intuitively too naive. Improving it with masking only the important things make intuitively sense.

**Reasons To Reject:**

1. There are still other methods that are trying to improve the domain-adaptive pre-training (e.g., [1]), which the proposed method should also consider compared with. The author should compare the pros and cons of changing the data (like masking) with changing the neurons (like [1])
2. Since the proposed method only masks the difference, will the model forget about the general knowledge in the LM? How the proposed method address this? Intuitively, a good pre-training objective should be able to consider both the shared and the different.


[1]: Adapting a Language Model While Preserving its General Knowledge, Ke et al., EMNLP 2022



**Reproducibility:**

4: Could mostly reproduce the results, but there may be some variation because of sample variance or minor variations in their interpretation of the protocol or method.

**Reviewer Confidence:**

1: Not my area, or paper was hard for me to understand. My evaluation is just an educated guess.

---

> ### Author Rebuttal · Authors · 2023-08-28
>
> Thank you for your insightful feedback and for recognizing the significance of our domain-adaptive pre-training approach. We've always believed that there's room to improve upon traditional masked language modeling, and that this line of research could be helpful in domains that pretrained models struggle with. We're grateful to see that you appreciate our effort in refining the process! Looking forward to addressing your comments below.
>
> — “There are still other methods that are trying to improve the domain-adaptive pre-training (e.g., [6]), which the proposed method should also consider compared with. The author should compare the pros and cons of changing the data (like masking) with changing the neurons (like [6])” ---
>
> This is an interesting baseline to run! We considered baselines in our paper that modified the masking strategy, but did not consider baselines that changed the neurons. We’ve implemented this paper, and found that our approach outperforms it by a wide margin: On ACL-ARC (where the metric used is Macro F1 [6] ), DGA scored 67.20, whereas Difference-Masking outperformed it by over 6% absolute to 74.04. On ChemProt (where the metric is accuracy [6]), Difference-Masking outperformed DGA as well: DGA scored 70.67 and Difference-Masking performed at 83.94 – an over 13% absolute increase. We ran these results over 5 trials, and are happy to incorporate the results into the paper. In their original paper, DGA performed continued pretraining on a much larger corpus; it is possible that their poor performance is due to their method’s reliance on a larger amount of information, whereas Difference-Masking performs well in the low-resource setting. We plan to include this in the next version of our paper.
>
> --- “Since the proposed method only masks the difference, will the model forget about the general knowledge in the LM? How the proposed method address this? Intuitively, a good pre-training objective should be able to consider both the shared and the different. ---
>
> This is another interesting point. It may be the case that the model will sacrifice some of the general knowledge in order to perform well on the downstream task. We agree that in some cases a good pretraining objective would balance the shared and the different, but we would argue that there are tasks for which this line of research would be very useful – where we don't need to preserve generality, because our primary concern is that the model perform well on the downstream task. For example, in the clinical domain it is critical that models perform strongly, even if this comes at a cost to its capabilities in another field – [2] implements this. In fact, some approaches such as [4] (also in the clinical domain) go as far as to pretrain from scratch in the target domain so achieve performance gains – we do not pretrain from scratch because the datasets are relatively small and to scrape large enough amounts of data from these domains to perform an effective pretraining (see [5]) would be computationally infeasible. Our method is dramatically less computationally and data expensive than a full domain-specific pretraining, leveraging the models' pretrainings to transfer to the target domain with relatively little compute.
>
> --
>
> [1] Gururangan, Suchin, et al. "Don't stop pretraining: Adapt language models to domains and tasks." arXiv preprint arXiv:2004.10964 (2020).
> [2] Lee, Jinhyuk, et al. "BioBERT: a pre-trained biomedical language representation model for biomedical text mining." Bioinformatics 36.4 (2020): 1234-1240.
> [3] Tuhin Chakrabarty, Christopher Hidey, and Kathy McKeown. 2019. IMHO fine-tuning improves claim detection. In NAACL.
> [4] Kexin Huang, Jaan Altosaar, and Rajesh Ranganath. 2019. ClinicalBERT: Modeling clinical notes and predicting hospital readmission. arXiv:1904.05342.
> [5] Kaplan, Jared, et al. "Scaling laws for neural language models." arXiv preprint arXiv:2001.08361 (2020).
> [6] Adapting a Language Model While Preserving its General Knowledge, Ke et al., EMNLP 2022

---

### Official Review · Reviewer_LPF7 · 2023-08-05

**Soundness:** 3

**Excitement:**

3: Ambivalent: It has merits (e.g., it reports state-of-the-art results, the idea is nice), but there are key weaknesses (e.g., it describes incremental work), and it can significantly benefit from another round of revision. However, I won't object to accepting it if my co-reviewers champion it.

**Paper Topic And Main Contributions:**

This paper introduces a method called Difference-Masking for continued pretraining. The approach prioritizes masking concepts that differentiate the target domain from the pretraining domain. It achieves this by selecting anchor words/regions relevant to the continued pretraining tasks and masking similar concepts during the process. The experimental results on language-only benchmarks and multimodal benchmarks demonstrate the effectiveness of this masking strategy for continued pretraining tasks.

**Questions For The Authors:**

Please see Reasons To Reject.


=====Post-rebuttal:

I appreciate the authors' diligence in their rebuttal, which partially addressed my concern. The concept of Difference-Masking continues to be an elegant and effective strategy, as convincingly demonstrated by the authors.

However, there are a few points that I believe still warrant further clarification or consideration in the final version:
1. About Distribution: While I acknowledge the authors' efforts in addressing the drawback to store the pretraining data, I still recommend a deeper exploration. The rebuttal discussed the use of distributions from a large dataset as a proxy for the pretraining distribution. To enhance the paper's robustness, I suggest a clearer distinction between using approximations and exact pretraining dataset statistics.

2. Masking Fairness in Video/Image Patching: the authors explained that the masking selection for patches is based on object region, while random masking is based on patch level -- this is somehow unfair. The authors may: (a) authors may use their strategy to conduct patch selection for masking, or (2)  compare their method to randomly mask the patches for random objects.

**Reasons To Accept:**

- The paper's writing is clear, and the proposed method is straightforward and easy to understand
- The performance of Difference-Masking on downstream language-only (ACL-ARC, ChemProt) and multimodal (Social-IQ, TVQA) tasks surpasses strong baselines, and the analysis provided is detailed.

**Reasons To Reject:**

- The solution's suboptimal aspect arises from the need to store the entire pretraining corpus when selecting anchors for continued pretraining, making it impractical. Other masking strategies do not require such storage.
- The extension of Difference-Masking to the multimodal setting is limited. The authors rely on labels/bboxes for anchor and mask selection, which might not be available for other multimodal tasks. The application of Difference-Masking to methods that patchify image/video input instead of extracting object regions is unclear.
- Some hyperparameters appear to be sensitive. The analysis in Figure 4 shows that even within the small range [15, 30], different numbers of anchors can lead to considerable performance differences, e.g., for ACL-ARC when K=15 and K=20.

**Reproducibility:**

4: Could mostly reproduce the results, but there may be some variation because of sample variance or minor variations in their interpretation of the protocol or method.

**Reviewer Confidence:**

4: Quite sure. I tried to check the important points carefully. It's unlikely, though conceivable, that I missed something that should affect my ratings.

---

> ### Author Rebuttal · Authors · 2023-08-28
>
> Thank you for taking the time to review our manuscript. We appreciate your positive feedback on the clarity of the paper's writing and the straightforwardness of our method. It is heartening to see that you recognize Difference-Masking's strong performance across both language-only and multimodal tasks, and found the analysis detailed.
>
> To address your concerns:
>
> –-- "The solution's suboptimal aspect arises from the need to store the entire pretraining corpus when selecting anchors for continued pretraining, making it impractical. Other masking strategies do not require such storage." –--
>
> We should perhaps clarify a misconception. We don't actually store the entire pretraining corpus for anchor selection. Instead, we determine the anchor "term frequency" based on the continued pretraining corpus. This corpus is notably smaller than the primary pretraining corpus, and we can't get around storing the continued pretraining corpus because it is required for continued pretraining.
>
> For the "inverse corpus frequency," we see where the potential confusion arises. While determining this value directly would require the entire pretraining corpus, we've leveraged ingenuity from prior works and approximated approximated large pretraining corpora using word distributions from vast online data sources. A notable example is the study in [1] that employs the Google Books dataset to approximate a large pretraining distribution. In line with this approach, we've turned to the Google Web Trillion Word Corpus to get our approximation.
>
> The beauty of our method lies in its simplicity: we just need word counts from our smaller, continued pretraining corpus and an aggregate measure that's already been meticulously worked out by others who've delved deep into word distributions on the web. We'd like to direct you to Section 3.4 where we've discussed this in detail. And of course, if there's any point you'd like us to elaborate on, we'd be happy to!
>
> --- "The extension of Difference-Masking to the multimodal setting is limited. The authors rely on labels/bboxes for anchor and mask selection, which might not be available for other multimodal tasks. The application of Difference-Masking to methods that patchify image/video input instead of extracting object regions is unclear." –--
>
> This is an interesting point, and one that warrants a deeper dive here and in our paper. The key contribution of Difference-Masking is to mask what makes the domain unique. We find that masking at a higher level of semantic abstraction is useful – in Appendix C we find that Difference-Masking applied at the word level is better than Difference-Masking applied at the token-level, and we find the same for Video Tokens at the bounding box level vs the patch level in Appendix B. There are many ways to consider "higher level semantic entities": for language we move from tokens to words, for vision we move from patches to bounding boxes, for audio we could move from raw audio to audio separated by word-boundaries...etc.
>
> We'd like to respectfully challenge one particular point: "The application of Difference-Masking to methods that patchify image/video input instead of extracting object regions is unclear." Our multimodal experiments use a vision backbone that actually does patchify image/video input. Our method calculates the probability of masking a patch based on the bounding box from which that patch is drawn. This principle is adaptable and not specific to vision. For example, we could imagine an extension to another modality where a semantically meaningful "object" is defined, for example audio by figuring out the corresponding word for a given raw audio token. We refer you to Appendix B and the MERLOT-Reserve paper [2] for details on the patching & masking process.
>
> Additionally, we could imagine applying this method based on multimodal embedding similarity, for example taking similarity between vision patches and text tokens, as determined by a multimodal model such as CLIP. In early experiments, we found that this did not perform nearly as well as our approach based on bounding boxes, perhaps because of CLIP’s difficulty with multimodal compositionality [3].
>
> –--"Some hyperparameters appear to be sensitive. The analysis in Figure 4 shows that even within the small range [15, 30], different numbers of anchors can lead to considerable performance differences, e.g., for ACL-ARC when K=15 and K=20."–--
>
> Our method does appear to be sensitive to the setting of K; we've investigated further and would like to present a nice intuition we’ve found for why we see the upside-down-U shaped graph in Figure 4. When the number of anchors K is equal to the number of unique words in the dataset, our method should converge to the random masking strategy. This is because when that happens, every word will have perfect similarity to a nearby anchor (itself), so each word will be equally weighted, defaulting to a random uniform masking probability. When the number of anchors K is very small (e.g., 1), we're adding a strong inductive bias into the masking strategy based on an aggregate measure of the dataset determining which word is most unique to dataset.
>
> So searching for the optimal the number of anchors K is meaningfully a question of coverage: how close are anchors in embedding space to different words that we'd like to mask? This can be roughly approximated by the variance of the anchors, and we've run an additional analysis and found that performance on the different datasets is actually tightly correlated with the variance of the anchor embeddings: using Pearson's Correlation Coefficient, ChemProt is .8463 (meaning that as the variance of the anchors increases the performance increases) and for ACL-ARC the correlation is -.9576 (as the variance of the anchors increases, the performance decreases). These are interesting results, and we actually see this as a benefit of our approach because it allows us to nicely calibrate the masking strategy to the domain, and to learn something about the domain in the process. For example, it seems that ACL-ARC benefits from a masking strategy that is more tightly coupled in the embedding space, whereas ChemProt benefits from a higher variance masking strategy. As this is the only hyperparameter of our method, and our experiments only take a few hours to run, it seems worthwhile for future work to search through to understand. We are happy to include these results in our paper as well.
>
> –
>
> Thank you for these insightful contributions to our work, and for engaging with it so carefully! We appreciated your comments and are happy to make these changes.
>
> [1] Liu, Fangyu, et al. "Do ever larger octopi still amplify reporting biases? Evidence from judgments of typical colour." arXiv preprint arXiv:2209.12786 (2022).
> [2] Zellers, Rowan, et al. "Merlot reserve: Neural script knowledge through vision and language and sound." Proceedings of the IEEE/CVF Conference on Computer Vision and Pattern Recognition. 2022.
> [3] Kamath, Amita, Jack Hessel, and Kai-Wei Chang. "Text encoders are performance bottlenecks in contrastive vision-language models." arXiv preprint arXiv:2305.14897 (2023).

---

### Meta-Review · Area_Chair_XkH7 · 2023-09-18

**Recommendation:** 4

**Metareview:**

The paper presents a simple and elegant modification to masking for domain adaptation in continued pre-training by focusing on masking tokens that are different in the new domain compared to the old domain. The authors also provide good experimental results, outperforming more complicated baselines on two NLP datasets (ACL-ARC, ChemProt) and two multimodal datasets (Social-IQ, TVQA).

The authors have addressed the reviewer concerns in the comments (clarifications for Reviewer LPF7), results on an additional baseline (Reviewer Yfzq) and additional experiments + clarifications for Reviewer o2x7.

---

### Decision · Program_Chairs · 2023-10-07

**Decision:**

Accept-Findings

**Comment:**

The paper presents a simple and elegant modification to masking for domain adaptation in continued pre-training by focusing on masking tokens that are different in the new domain compared to the old domain. The authors also provide good experimental results, outperforming more complicated baselines on two NLP datasets (ACL-ARC, ChemProt) and two multimodal datasets (Social-IQ, TVQA).

The authors have addressed the reviewer concerns in the comments (clarifications for Reviewer LPF7), results on an additional baseline (Reviewer Yfzq) and additional experiments + clarifications for Reviewer o2x7.